# A numerical model of the MICP multi-process considering the scale size

**Xianxian Zhu[1], Jianhua Wang[1], Haili Wang[2], Yujie Li [3]***

**1** Zhejiang Qiantang River Basin Center, Hangzhou, China, **2** Survey and Design Institute of Qiantang River Administration Bureau of Zhejiang Province, Hangzhou, China, **3** Ocean College, Zhejiang University, Zhoushan, China

\* liyujies@163.com

**Data Availability Statement:** All relevant data are within the manuscript and its Supporting Information files.

**Funding:** The authors would like to acknowledge the supports by the Zhejiang Water Conservancy

## Abstract

As an environmentally friendly and controllable technology, Microbially induced carbonate precipitation (MICP) has broad applications in geotechnical and environmental fields. However, the longitudinal dispersivity in MICP multi-process varies with the scale size. Ignoring the effect of the scale size of the research object on the dispersivity leads to the inaccuracy between the numerical model and the experiment data. Thus, this paper has established the relationship between the scale size and the dispersivity initially, and optimized the theoretical system of MICP multi-process reaction. When scale size increases logarithmically from $10^{-2}$ m to $10^5$ m, longitudinal dispersivity shows a trend of increasing from $10^{-3}$ m to $10^4$ m. The distribution of calcium carbonate is closer to the experimentally measured value when the size effect is considered. After considering the scale size, the suspended bacteria and attached bacteria are higher than the cased without considering the size effect, which leads to a higher calcium carbonate content. Scale has little effect on the penetration law of the suspended bacteria. The maximum carbonate content increases with the increase of the initial porosity, and the average carbonate shows a significant increasing trend with the increase of the bacterial injecting rate. In the simulation of the microbial mineralization kinetic model, it is recommended to consider the influence of the scale size on the MICP multi-process.

## 1. Introduction

As shown in Fig 1, microbially induced carbonate precipitation (MICP) has attracted extensive attention in coastal zone reinforcement [1–3], erosion and scour protection [4–7], ground improvement [8, 9] and pile foundation reinforcement [10, 11]. Among them, urea is decomposed by *Sporosarcina pasteurii* into ammonium and calcium, which combines with carbonate ion to produce calcium carbonate, leading to the particle coating, particle cementing, pore filling in sand [12–23]. The chemical reaction is shown as below.

$$CO(NH_2)_2 + 2H_2O \xrightarrow{bacteria/urease} 2NH_4^+ + CO_3^{2-} \tag{1}$$

$$CO_3^{2-} + Ca^{2+} \rightarrow CaCO_3 \downarrow \tag{2}$$

Key Science and Technology Project (RB2120), Hainan Special PhD Scientific Research Foundation of Sanya Yazhou Bay Science and Technology City (HSPHDSRF-2022-04-002). There was no additional external funding received for this study. The funders had no role in study design, data collection and analysis, decision to publish, or preparation of the manuscript.

**Competing interests:** The authors have declared that no competing interests exist.

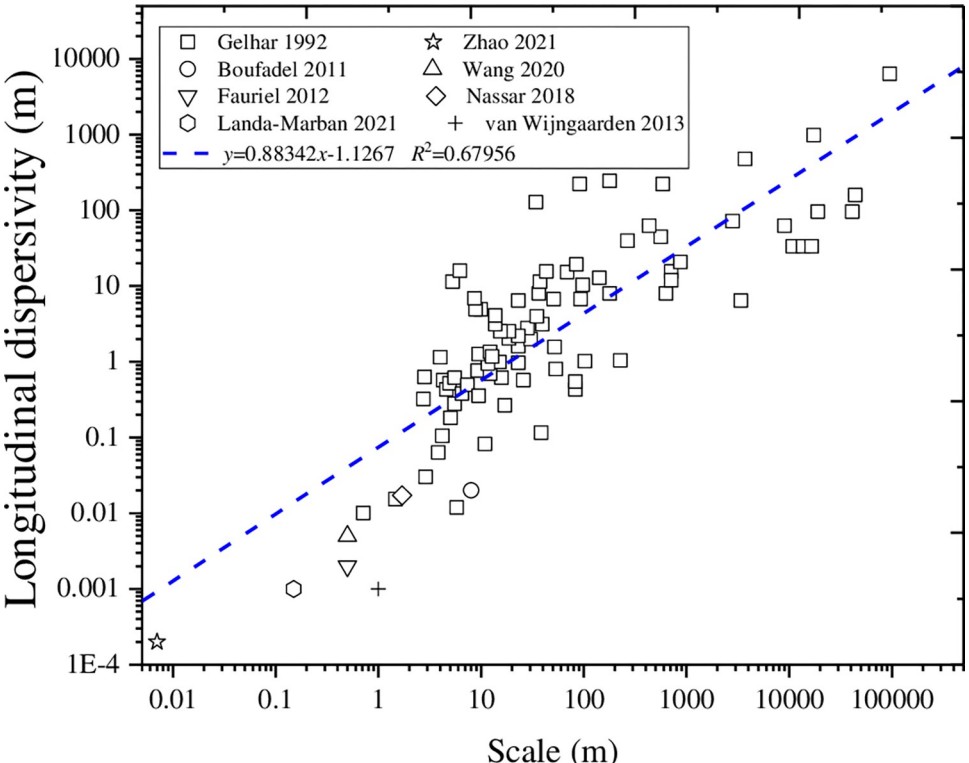

**Fig 1. Longitudinal dispersivity versus of the scale size in saturated porous media.**

The MICP reaction in sand is particularly complex process, involving multiple processes under the coupling of physical-chemical-biological fields. The injecting rate directly affects the progress of the MICP reaction. The flow velocity distribution of pore fluid in sand mainly includes three forms: 1) the uneven flow velocity distribution in single pore, the maximum flow velocity appears in the middle of the pore due to the influence of particle friction; 2) Split flow in multiple pores, the difference in pore size and shape leads to the difference in flow velocity; 3) Lateral dispersion caused by particle blocking, which is the main factor causing the lateral dispersion of the solute. Importantly, when the MICP reaction occurs in the sand, the deposited calcium carbonate encapsulates the particles, plugs pores, and connects the particles, which all alter the flow velocity distribution of the pore. The pore diameter decreases when calcium carbonate is deposited on the particle surface, resulting in an increase in pore velocity. In addition, the solute (urea, calcium, ammonium and bacteria) is transported over long distances along the flow direction, while the diffusion still appears perpendicular to the direction of the flow. As a result, the concentration of the solute exhibits a non-uniform distribution characteristic similar to an ellipse over time.

Up to now, most of the research about MICP has focused on the laboratory experiments. The spatiotemporal evolutions of various chemical substances in MICP multi-process cannot be obtained in real time. The field experiments cost a lot of manpower and resources, and are highly dependent on the field conditions. Therefore, reliable model is of great significance for understanding biochemical processes in MICP reaction [20, 24, 25].

The convection-diffusion-reaction are always used to describe the evolutions of the suspended bacteria, urea, ammonium, and calcium in MICP numerical model. Hydrodynamic dispersion is the main way of the substance migration in sand, including the molecular

diffusion caused by concentration gradients and the mechanical dispersivity caused by pore flows. Dispersion is significantly directional, dependent on pore characteristics and scale size. According to Gelhar (1992), the longitudinal dispersivity, which is the most difficult parameter to determine in solute transport, ranged from $10^{-2}$ to $10^4$ m for size ranging from $10^{-1}$ to $10^5$ m [26]. This value cannot be directly measured. The results of the Toride's (2003) study show that the dispersivity does not change with the velocity of the pore flow [27]. Godoy (2018) found that the longitudinal dispersivity have significant size effects with assuming a liner relationship between the longitudinal dispersivity and the flow velocity [28]. However, the hydrodynamic dispersivity coefficient was mostly taken as the molecular dispersivity coefficient in the simulation of MICP multi-process, the mechanical dispersivity and the effect of the pore tortuosity on the molecular dispersivity were always ignored [23, 29, 30]. Some scholars have divided hydrodynamic dispersivity into mechanical dispersivity and molecular dispersivity in a more detailed way, but the values of dispersivity are quite different [25, 31–33]. The scale of the research objects for the MICP technology ranges from micro to macro, but the influence of the scale size on the dispersivity in MICP multi-process has not yet been reported. There is no unified way for the value of the hydrodynamic dispersivity in the numerical model of the MICP multi-process, and improper handling leads to the inaccuracy between the model and the experimental data.

Thus, this study established the relationship between the scale size and the longitudinal dispersivity initially, and optimized the MICP multi-process numerical model. Ignoring the effect of the scale size of the research object on the dispersivity leads to the inaccuracy between the numerical model and the experiment data. The influences of the scale size on the dispersivity, bacteria and calcium carbonate distribution in MICP multi-process were analysed. In addition, the parametric sensitivity analysis was also performed for initial porosity and bacterial injecting rate.

## 2. Mathematical model of the MICP multi-process

### 2.1 Multi-stages of MICP grouting

Biomineralization in sand involves multi-process of physical-chemical-biological reaction. It can be divided into: 1) Bacteria injection stage (Phase B), the convection and diffusion of the suspended bacteria appear in sand, and the attached bacteria accumulate on the particle surface; 2) bacteria retention stage (Phase R), bacteria show molecular diffusion and adsorption; 3) Cementing solution injection stage (Phase C), this stage includes the convective diffusion reaction of all solutes, the formation of the calcium carbonate, the reduction in porosity and the permeability of the sand. Due to the consumption of the substances and the encapsulation of the calcium carbonate, the decay of the bacteria is considered throughout the process.

### 2.2 The bacterial behaviour

Bacteria shows the growth, decay and attachment in sand. Due to the interception and scree of particles, some bacteria adhere to the particle surface ($C_{bacs}$), while some bacteria are free in the pore and migrate with the fluid ($C_{bacl}$). The number and activity of the bacteria gradually weakens because of the lack of the nutrient supply and the encapsulation of the calcium carbonate. In general, the first-order kinetic equation is used to describe the behaviour of bacteria in sand [25]. The equations are shown below.

$$\frac{\partial C_{bact}}{\partial t} = -k_{\mathrm{d}} C_{bact} \tag{3}$$

$$\frac{\partial C_{bacs}}{\partial t} = k_{att} C_{bacl} - k_d C_{bacs} \tag{4}$$

where $k_d$ is a constant decay rate, and $k_{att}$ is the constant attachment rate. The detail is shown in Li et al., (2023) [34], the author established the kinetic theory of biomineralization in seabed under the action of waves, and this paper focuses on analysing the influence of scale size on the biomineralization reaction in sand.

## 2.3 The fluid equations and solute transport diffusion equations

Considering that the fluid is incompressible, the continuity equation is used to describe the relationship between fluid pressure and flow velocity (Eq 5). It is worth noting that the fluid density changes during the mineralization reaction, as shown in Eq 6.

$$\frac{d(\varphi \rho_l)}{dt} = -\nabla \cdot (\rho_l q) - m_{urea} \varphi k_{rea} - m_{Ca^{2+}} \varphi k_{rea} + 2 m_{NH_4^+} \varphi k_{rea} \tag{5}$$

where $\varphi$, $m_{urea}$, $m_{Ca^{2+}}$, $m_{NH_4^+}$ represent the porosity, molar mass of urea, calcium and ammonium, respectively. $k_{rea}$ is the urea hydrolysis constant, determined by Eq 15.

$$\rho_l = \rho_w + 0.0154994 kg/mol \cdot C_{urea} + 0.0867338 kg/mol \cdot C_{Ca^{2+}} + 0.0158991 kg/mol \cdot C_{NH_4^+} \tag{6}$$

where $\rho_w$, $C_{urea}$, $C_{Ca^{2+}}$, $C_{NH_4^+}$ are the water density, substance concentration of urea, calcium and ammonium, respectively.

Assuming that the system is laminar flow conditions, the relationship between the average velocity $q$ and liquid pressure $p_l$ can be described by Darcy's law (Eq 6).

$$q = -\frac{K}{\mu_l} \cdot (\nabla p_l + \rho_l \boldsymbol{g}) \tag{7}$$

where $K$ is the matrix permeability, $\boldsymbol{g}$ is the gravity acceleration, the $\mu_l$ and $\rho_l$ represent the liquid viscosity and the density.

The general form of the convective-diffusion-reaction is as follows:

$$\varphi \frac{\partial C_{urea}}{\partial t} = \nabla \cdot (\varphi \boldsymbol{D}^* \cdot \nabla C_{urea}) - q \cdot \nabla C_{urea} - \varphi k_{rea} \tag{8}$$

$$\varphi \frac{\partial C_{NH_4^+}}{\partial t} = \nabla \cdot \left(\varphi \boldsymbol{D}^* \cdot \nabla C_{NH_4^+}\right) - q \cdot C_{NH_4^+} + 2\varphi k_{rea} \tag{9}$$

$$\varphi \frac{\partial C_{Ca^{2+}}}{\partial t} = \nabla \cdot (\varphi \boldsymbol{D}^* \cdot \nabla C_{Ca^{2+}}) - q \cdot C_{Ca^{2+}} - \varphi k_{rea} \tag{10}$$

$$\varphi \frac{\partial C_{bacl}}{\partial t} = \nabla \cdot (\varphi \boldsymbol{D}^* \cdot \nabla C_{bacl}) - q \cdot C_{bacl} - \varphi k_d C_{bacl} - \varphi k_{att} C_{bacl} \tag{11}$$

where $C_i$ represents the concentration of the urea, calcium, ammonium and suspended bacteria, $\boldsymbol{D}^*$ represent the hydrodynamic dispersivity coefficient in sand. Solute diffusion in saturated sand mainly includes mechanical diffusion and molecular diffusion [35], so there are:

$$\boldsymbol{D}^* = (\alpha_L - \alpha_T) \frac{v \otimes v}{|v|} + \alpha_T |v| \boldsymbol{I} + \frac{\varphi}{\tau} \boldsymbol{D} \tag{12}$$

where, $D^*$ and $D$ represent the hydrodynamic dispersivity coefficient in sand and water, $\alpha_L$ and $\alpha_T$ are the vertical and horizontal dispersivity, $v$ is the pore water velocity $(q/\varphi)$ and $\tau$ represents the tortuosity, the Millington-Quirk model is used to describe the tortuosity in this paper.

$$\tau = \varphi^{-1/3} \tag{13}$$

When the research object is one-dimensional condition, Eq 12 degenerates into:

$$\boldsymbol{D}^* = \alpha_L \cdot v + \frac{\varphi}{\tau}\boldsymbol{D} \tag{14}$$

The value of the hydrodynamic dispersivity coefficient $D$ is taken as 2e-9 m$^2$/s in this paper. According to Eq 1, consuming 1 mol urea will produce 2 mol calcium in urea hydrolysis. The research of Whiffin (2004) show that the ammonium has no obvious inhibition on urease activity [36]. Therefore, the Monod equation [24, 37] is used to describe the kinetics of urea hydrolysis.

$$k_{rea} = u_{sp}(C_{bacs} + C_{bacl}) \cdot \frac{C_{urea}}{C_{urea} + k_m} \cdot \exp\left(-\frac{t}{t_d}\right) \tag{15}$$

where $k_m$, $t_d$, $u_{sp}$ is the half-saturation constant when the reaction rate is reduced by 50%, the time constant and the maximum urease constant, respectively.

## 2.4 Calcium carbonate deposition and porosity evolution

Calcium carbonate produced by biomineralization deposit between the sand particles, fill pores, reduce the porosity and permeability of the sand [4, 13, 38–40]. Assuming that the deposited calcium carbonate does not transport in the reaction time, there are:

$$\frac{\partial \varphi}{\partial t} = -\frac{1}{\rho_c}m_{CaCO_3}\varphi k_{rea} \tag{16}$$

$$\frac{\partial C_{CaCO_3}}{\partial t} = m_{CaCO_3}\varphi k_{rea} \tag{17}$$

where $\rho_c$ is the density of the calcium carbonate. In addition, the Kozeny-Carman (KC) equation and the effective porosity are adapted in this paper according to Wang et al. 2020a [25], in which the enough details can be obtained. The boundary conditions at multi-process are shown in Table 1.

# 3. Results and analysis

## 3.1 The relationship between scale size and the longitudinal dispersivity

Dispersivity is a key parameter in the convection-diffusion, which characterizes the dispersion characteristics of solutes, and affected by scale size. The dispersivity measured indoors is often different from that measured in the field by one or several orders of magnitude, thus the results of the indoor test cannot be applied to the field. Some studies have found that the dispersivity increase with the increase of the scale size [26, 28, 41–43]. According to this theory, the dispersivity obtained indoors can be extrapolated to the field scale by scale analysis. However, it is still a worldwide problem due to the uncertainties in scale size. Thus, a large amount of data between longitudinal dispersivity and scale size are summarized in Fig 1 [20, 24–26, 32, 44–46]. It can be found that the longitudinal dispersivity in saturated porous media increase with

**Table 1. Boundary conditions of the MICP process.**

| Multi-process | Phase B | Phase R | Phase C |
|---|---|---|---|
| **Top** | | | |
| $q_1$ | $q_{in}$ | / | / |
| $p$ | / | $p_{atm}$ | $p_{atm}$ |
| $C_{bacl}$ | $C_{bact0}$ | $(\boldsymbol{D}^* \cdot \varphi\nabla C)\cdot \mathbf{n}=0$ | / |
| $C_{urea}$ | / | / | $(\boldsymbol{D}^* \cdot \varphi\nabla C)\cdot \mathbf{n}=0$ |
| $C_{Ca2+}$ | / | / | $(\boldsymbol{D}^* \cdot \varphi\nabla C)\cdot \mathbf{n}=0$ |
| $C_{NH4+}$ | / | / | $(\boldsymbol{D}^* \cdot \varphi\nabla C)\cdot \mathbf{n}=0$ |
| **Bottom** | | | |
| $q_2$ | / | / | $q_{in}$ |
| $p$ | $p_{atm}$ | $p_{atm}$ | / |
| $C_{bacl}$ | $(\boldsymbol{D}^* \cdot \varphi\nabla C)\cdot \mathbf{n}=0$ | $(\boldsymbol{D}^* \cdot \varphi\nabla C)\cdot \mathbf{n}=0$ | $(\boldsymbol{D}^* \cdot \varphi\nabla C)\cdot \mathbf{n}=0$ |
| $C_{urea}$ | / | / | $C_{urea0}$ |
| $C_{Ca2+}$ | / | / | $C_{Ca2+0}$ |
| $C_{NH4+}$ | / | / | $C_{NH4+0}$ |

the increase of the scale size. The relationship is as follows:

$$lg\alpha_L = 0.88342 lgL_0 - 1.1267 \tag{18}$$

where $L_0$ represents the scale size along the main flow direction. When scale size increases logarithmically from $10^{-2}$ m to $10^5$ m, longitudinal dispersivity also shows a trend of increasing from $10^{-3}$ m to $10^4$ m.

According to Fig 1, the dispersivity measured in the laboratory can be extended to field. Although it is not particularly accurate, it still has good reliability. The value of the longitudinal dispersivity at different scales can be calculated by Eq 18, and then the simulation of the transport and reaction of the corresponding substances can be carried out. Especially for the large-scale MICP multi-process, there is no report on the influence of the scale size on the calculation results.

## 3.2 Model calibration and verification

The results of the Martinez (2013) are used to calibrate and verify the model [47]. As shown in Table 2, Test 1B is used for calibrating. The parameters attachment rate $k_{att}$ and maximum urease constant $u_{sp}$ were determined according to Test 1B. And the Test 2B and Test 3A are used for verification. According to the Fig 2(A), although the value of the penetration curve of the suspended bacteria is far from the measured value at the time of 1h, the numerical model can able to reflect the penetration law of the suspended bacteria. The calcium carbonate profile is in high agreement with the measured value, and the permeability coefficient after the MICP reaction is also similar. The permeability coefficient measured experimentally is $4.5\times10^{-4}$ cm/s, while the calculated value is $4.11\times10^{-4}$ cm/s. All calibration parameters are summarised in Table 3.

The verification results are shown in Fig 3. It can be found that when considering the size effect, the concentration of attached bacteria is higher than that without considering the size effect (Fig 3A), leading to a higher calcium carbonate content, which is more in line with the calcium carbonate content measured in the experiment (Fig 3B). However, it has little effect on the penetration law of the suspended bacteria (Fig 3C and 3D). From Fig 3(E), the concentration of the suspend bacteria is higher than that without considering the size effect in

**Table 2. Treatment steps of calibration and verification case.**

| Test No. | Test1B(calibration) | Test2B(verification) | Test3A(verification) |
|---|---|---|---|
| Sand sample length (cm) | 50 | 50 | 50 |
| Initial porosity | 0.37 | 0.35 | 0.36 |
| Initial perimeability (cm/s) | 3e-2 | 3.5e-3 | 2.2e-2 |
| **Phase B** | | | |
| Flow type | Continous | Continous | Continous |
| Injecting rate (mL/min) | 10 | 10 | 10 |
| Source | top | bottom | top |
| Initial cell concentration (cells/mL) | 7.2e5 | 2.5e5 | 4.3e5 |
| Volume of cell solution (mL) | 442.1 | 393.9 | 598.4 |
| **The Time of Phase R (h)** | 8.3 | 12 | 8 |
| **Phase C** | | | |
| Flow type | Continous | Continous | Stop |
| Injecting rate (mL/min) | 2.2 | 4.4 | 10 |
| Source | bottom | bottom | bottom |
| $C_{urea}:C_{NH4+}:C_{Ca2+}$ | 0.3:0.3:0.1 | 0.3:0.3:0.1 | 0.05:0.05:0.05 |
| Injecting time (h) | 81.267 | 48.769 | 2h/pulse |

Test3A, the distribution characteristics of the suspend bacteria are close to the measured values when the size effect is considered. Fig 3(F) shows the evolution of the urea, calcium and ammonium over the time in Test 3A. After considering the size effect, the consumption of reactants (urea and calcium) at the same time is larger than that in the case which the size effect is not considered. Correspondingly, the product (ammonium) is also higher.

In summary, this numerical model can reflect the kinetics of biomineralization and the law of calcium carbonate precipitation, the calculated results are closer to the experimental data after taking the scale size into account. Table 4 shows the design of calculation cases, the effects of scale size $L_0$, bacterial injecting rate $q_1$, and the initial porosity $\varphi_0$ on the MICP multi-process were studied systematically.

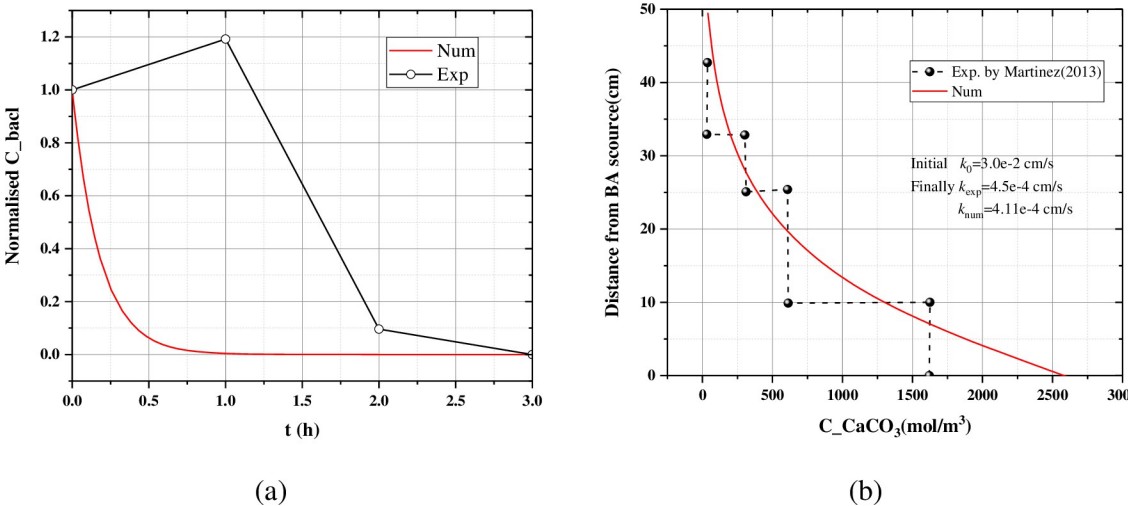

(a)                                              (b)

**Fig 2.** The results of the parameter calibration: (a) Penetration curves of the suspended bacteria; (b) Axial profile of the measured carbonate and the numerical carbonate of Test1B.

**Table 3. Model parameters.**

| Model parameters | Value | Reference |
|---|---|---|
| Water density at 25˚C $\rho_w$ ($kg/m^3$) | 1000 | |
| Viscosity at 25˚C $\mu_l$ ($Pa \cdot s$) | 0.001 | |
| Constant $a$ | 0.33 | Wang (2020) [25] |
| Critical porosity $\varphi_c$ | 0.25 | estimated |
| Diffusion constant $D$ ($m^2/s$) in water phase | 2e-9 | Wang (2020) [25] |
| Attachment rate $k_{att}$ ($1/s$) | 1.52e-3 | estimated |
| Decay rate $k_d$ ($1/s$) | 3.18e-7 | Wang (2020) [25] |
| Half saturation constant $k_m$ ($mol/m^3$) | 10 | Van Paassen(2009) [48] |
| Time constant $t_d$ ($s$) | 288000 | Fauriel(2012) [24] |
| Maximum urease constant $u_{sp}$ ($mol/m^3/s/cells/mL$) | 1.4e-8 | estimated |
| Molar mass $m_{urea}$, $m_{Ca2+}$, $m_{NH4+}$, $m_{CaCO3}$ ($kg/mol$) | 0.078,0.04,0.018,0.1 | |
| Calcaite density $\rho_c$ ($kg/m^3$) | 2710 | |

## 3.3 Effect of the scale size

Fig 4A–4c show the axial profile of the suspended bacteria with the different scale size at the end of the Phase B. After considering the scale size, the suspended bacteria are higher than that without considering the scale size. The point where the suspended bacteria is zero is defined as the critical point (Fig 4(B)), which indicates the penetration ability of the substance. In order to quantify the effect of the scale size, the error of the mean $\delta$ (%) is defined:

$$\delta = \frac{C_{i(scaleeffect)} - C_{i(original)}}{C_{i(original)}} \times 100\% \tag{19}$$

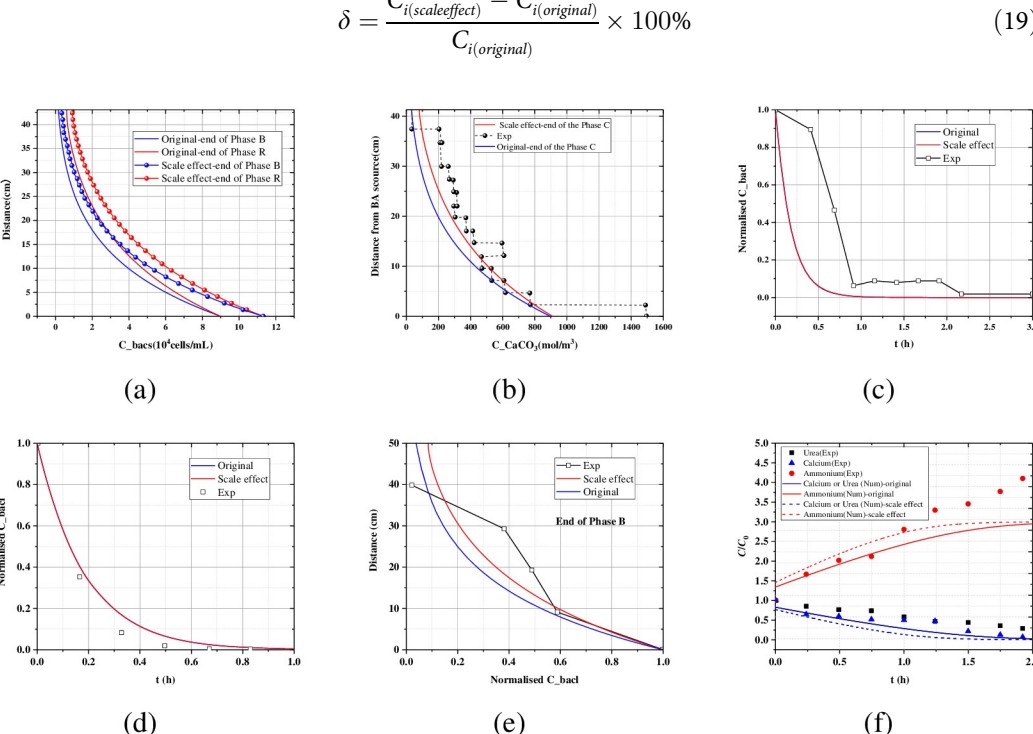

**Fig 3.** Axial profile of (a) the attached bacteria in Test2B; (b) the carbonate in Test2B; (c) Penetration curves of the suspended bacteria in Test2B; (d) Axial profile of the suspended bacteria in Test3A; (e) Penetration curves of the suspended bacteria Test3A; (f) Evolution of the urea, calcium and ammonium over time in Test3A.

**Table 4. Design of calculation cases.**

| Variables | Value | Unit |
|---|---|---|
| Initial bacterial concentration $C_{bact0}$ | 7.2e5 | cells/mL |
| Injecting rate of cell solution $q_1$ | 5,**10**,15,20,25 | mL/min |
| Injecting volume of the cell solution $V_c$ | 442.1 | mL |
| Initial perimeability coefficient $k_0$ | 3e-4 | m/s |
| Initila porosity $\varphi_0$ | 0.35,**0.37**,0.39,0.41,0.43 | |
| Scale size $L_0$ | 0.1,**0.5**,1,5,10 | m |
| The time interval of the PhaseR | 8.3 | h |
| $C_{urea}:C_{NH4+}:C_{Ca2+}$ | 0.3:0.3:0.1 | mol/L |
| Injecting rate of cementing solution $q_2$ | 2.2 | mL/min |
| Injecting time of cementing solution $t_2 - t_1$ | 81.267 | h |

The parameters of the standard calculated group are bold.

where $i$ include the suspended bacteria, the attached bacteria, urea, calcium, ammonium and carbonate. As shown in Fig 4(D), the mean error of the average suspended bacteria increases with the increase of the scale size. Fig 5A–5C show that the attached bacteria also showed similar laws to the suspended bacteria.

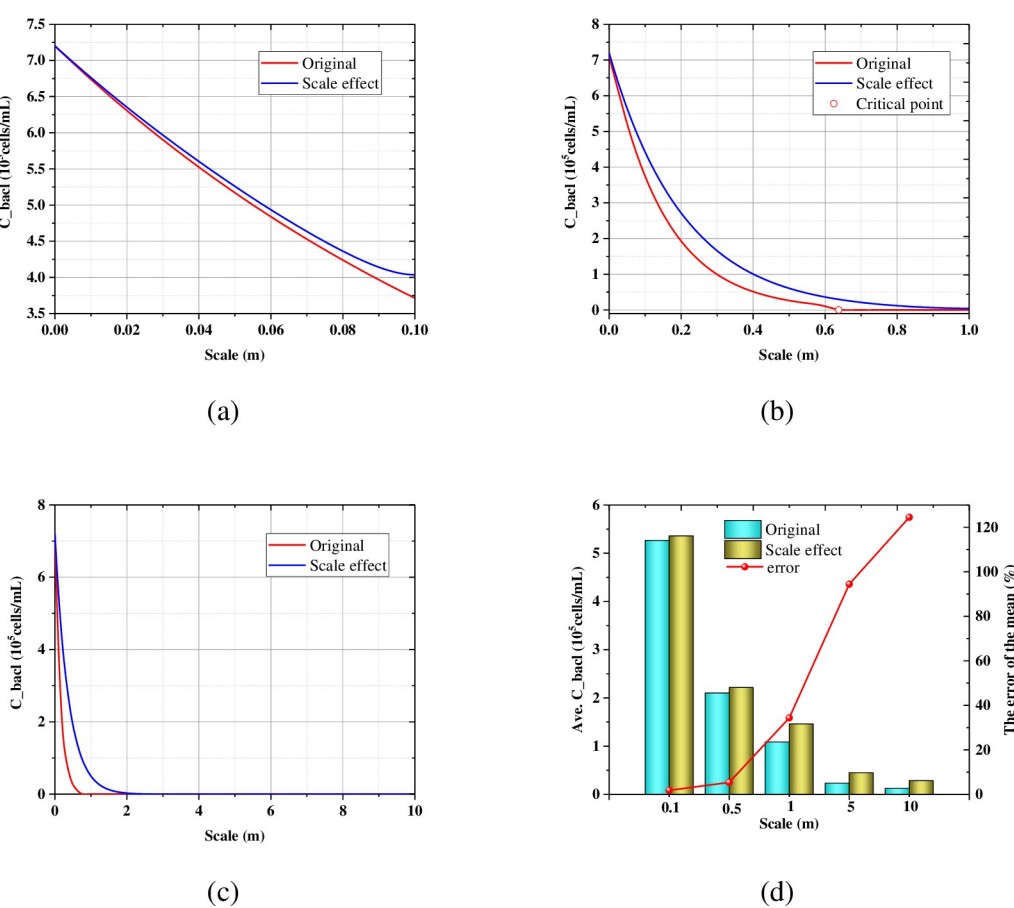

**Fig 4.** Axial profile of (a) the suspended bacteria in 0.1m; (b) the suspended bacteria in 1.0m; (c) the suspended bacteria in 10m at the end of the Phase B; (d) The influence of the scale size on the average suspended bacteria.

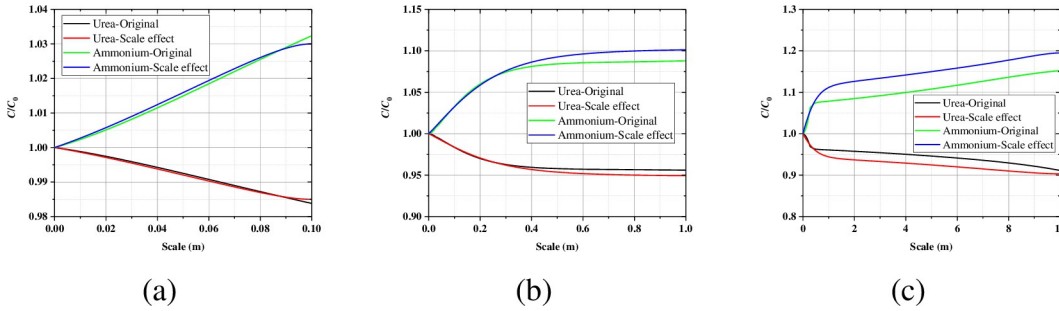

**Fig 5.** Axial profile of (a) the attached bacteria in 0.1m; (b) the attached bacteria in 1.0m; (c) the attached bacteria in 10m at the end of the Phase B; (d) The influence of the scale size on the average attached bacteria.

Fig 6 shows the axial profile of the concentration of the urea (calcium) and the ammonium at the end of the Phase C, when the sand column height is 0.1 m, considering the scale size has little effect on the concentration of urea and ammonium. When the sand column height

**Fig 6.** Axial profile of the urea and ammonium in (a) 0.1m; (b) 1.0m; (c) 10m at the end of the Phase C.

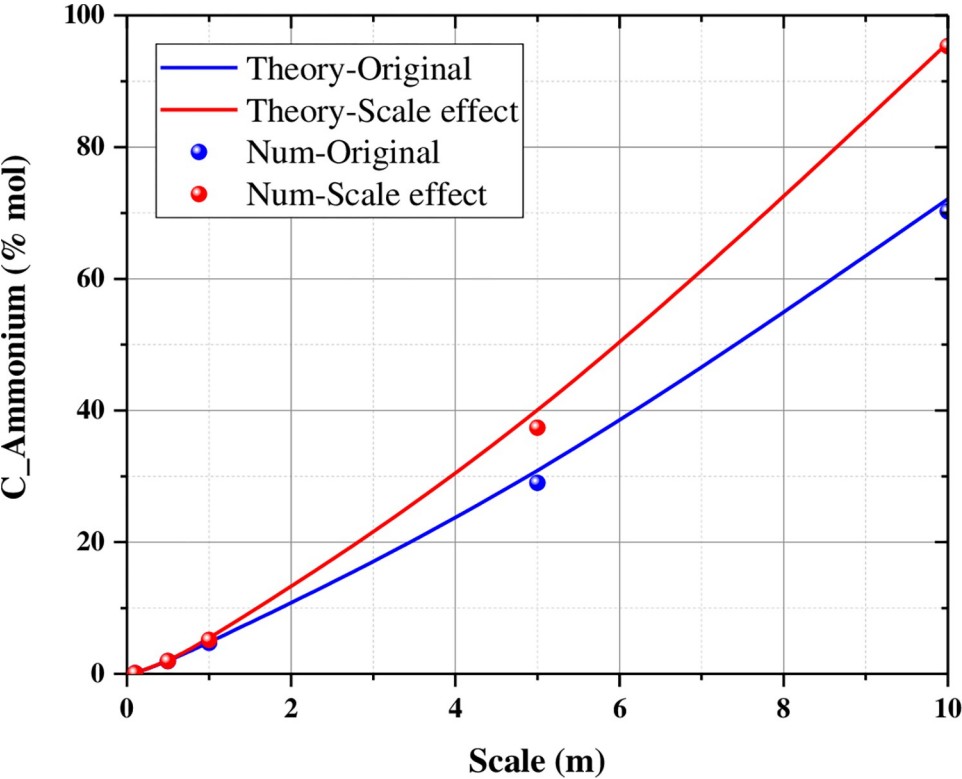

**Fig 7. The comparison between the theoretical ammonium production calculated by 100% conversion rate and the numerical ammonium production.**

increases to 1 m, more urea is consumed after considering the size effect, resulting in more ammonium ions. The influence after considering the scale size is more significant at the 10 m sand column height.

When the conversion rate is assumed to be 100%, the theoretical ammonium production (consumption of 1 mol urea to produce 2 mol ammonium) is highly consistent with the numerical ammonium production. And with the increase of scale size, the difference between the ammonium concentration after considering the size effect and the ammonium concentration without considering the size effect increases sequentially (Fig 7). The maximum value of the difference can be as high as 28% when the calculated size is 10 m in this paper.

Carbonate is the product of the MICP multi-process, which increases strength, stiffness, and reduces permeability and porosity of the sand. Fig 8A–8C show the axial profile of the carbonate with the different scale size, and Fig 8(D) gives the average carbonate value and the error. The effect of the scale size on carbonate is also similar to that of the suspended bacteria.

Table 5 summarized the location of the critical point of each substance in the MICP multi-process. When the scale size is less than 0.5m, all substances penetrate the entire calculation height (the penetration ratio is 100%). At the end of the Phase B, the penetration ratio of the $C_{bacl}$ for 0.1 m, 0.5 m, 1m, 5 m, 10 m scale size is 100%, 100%, 63.8%, 18.66%, 15.87% without considering size effect. While the value of considering the size effect is 100%, 100%, 100%, 60.7%, 39.68%, respectively. This suggests that ignoring the size effect underestimates penetration capability, which is detrimental to model accuracy and cost considerations. Both the penetration ratio of the $C_{bacs}$ at the end of Phase B and the carbonate at the end of the Phase C show a similar pattern of the $C_{bacl}$.

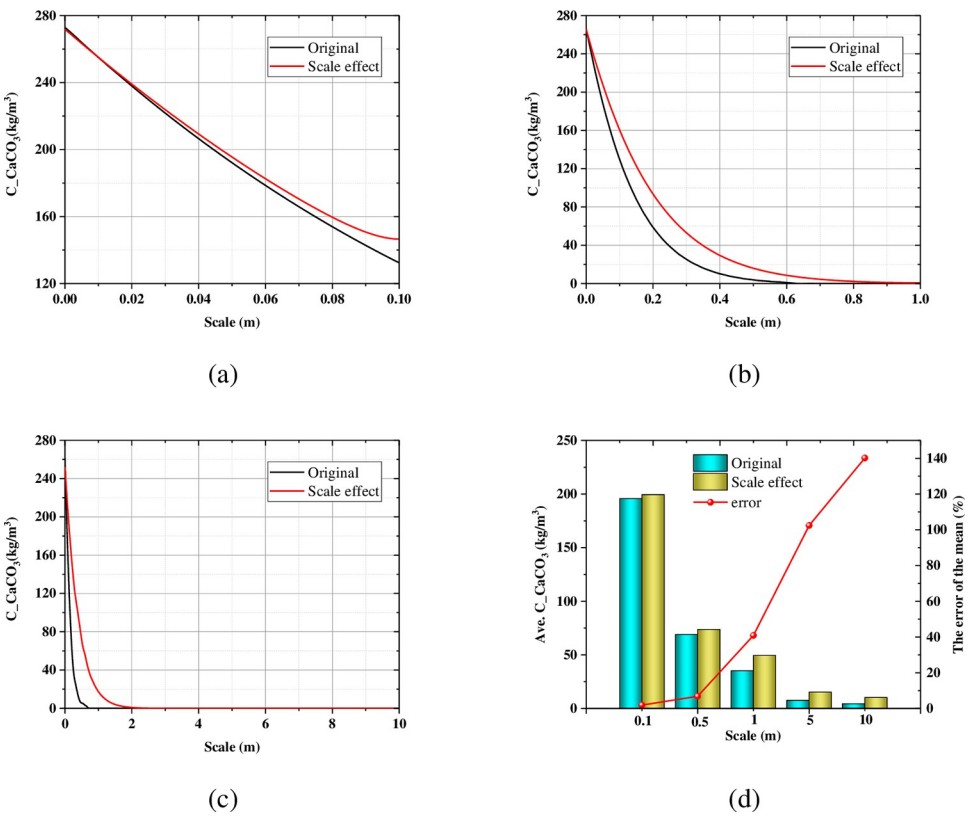

**Fig 8.** Axial profile of the carbonate in (a) 0.1m; (b) 1.0m; (c) 10m at the end of the Phase B; (d) The influence of the scale size on the average carbonate.

## 3.4 Effect of the porosity and the bacteria injecting rate

The porosity affects the convection diffusion reaction of the solute, so it is also necessary to pay attention to its effect on the MICP multi-process. Here, a 0.5 m-long model is used to conduct a study. From Fig 9A and 9B, under the condition of small initial porosity, there are more suspended and attached bacteria, which relies on the sieving effect of the sand on bacteria. But

**Table 5. Influence of the scale on the critical point location.**

| Scale size (m) | 0.1 | 0.5 | 1 | 5 | 10 |
|---|---|---|---|---|---|
| $C_{bacl}$ | | | | | |
| Original-end of Phase B | /*100 | /*100 | 0.638/*63.8 | 0.933/*18.66 | 1.587/*15.87 |
| Scale effect-end of Phase B | /*100 | /*100 | /*100 | 3.035/*60.7 | 3.968/*39.68 |
| $C_{bacs}$ | | | | | |
| Original-end of Phase B | /*100 | /*100 | 0.619/*61.9 | 0.654/*13.08 | 0.675/*6.75 |
| Scale effect-end of Phase B | /*100 | /*100 | /*100 | 2.936/*58.72 | 2.579/*25.79 |
| Original-end of Phase R | /*100 | /*100 | 0.631/*63.1 | 0.694/*13.88 | 0.754/*7.54 |
| Scale effect-end of Phase R | /*100 | /*100 | /*100 | 3.115/*62.3 | 2.698/*26.98 |
| $C_{CaCO3}$ | | | | | |
| Original-end of Phase C | /*100 | /*100 | 0.631/*63.1 | 0.694/*13.88 | 0.754/*7.54 |
| Scale effect-end of Phase C | /*100 | /*100 | /*100 | 1.448/*28.96 | 1.984/*19.84 |

*The penetration ratio, defined as a percentage of the ratio of the critical point location to the total height.

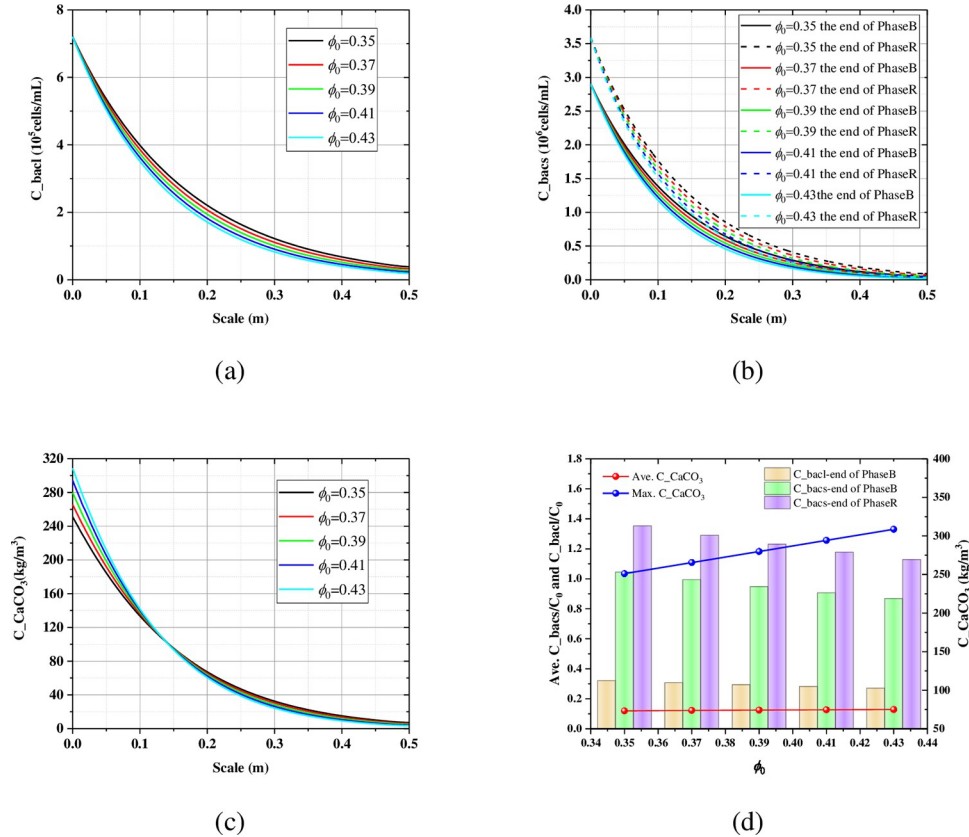

**Fig 9.** The influence of the initial porosity on the (a) axial profile of the suspended bacteria at the end of Phase B; (b) axial profile of the attached bacteria at the end of the Phase B and Phase C; (c) axial profile of the carbonate at the end of the Phase C; (d) the average bacteria and the carbonate.

the initial porosity has no effect on the maximum concentrations of the suspended and attached bacteria. It can be found from Fig 9(C) that small porosity is more conducive to the uniform distribution of carbonate. The initial porosity has less effect on the average carbonate. But the maximum carbonate content, which controls the permeability of the porous media, increases with the initial porosity (Fig 9D). The value of which is 251, 266, 280, 294, 309 kg/m³ for the initial porosity of 0.35, 0.37, 0.39, 0.41, 0.43.

As shown in Fig 10(A), the concentration and penetrating ability of the suspended bacteria increase with the increase of the bacteria injecting rate at the same injecting time. From Fig 10 (B), when the injecting rate is 5 mL/min, the attached bacteria still had a critical point in the height of 0.5 m, which indicated that the attached bacteria does not completely penetrate the sand column at this flow rate. When the injecting rate increase to 10 mL/min, the critical point of the attached bacteria disappears. And the concentration of the attached bacteria increases with the increase of the injecting rate. Fig 10(C) shows the spatial distribution of the carbonate with different injecting rates, which increase sequentially with the increase of the injecting rates. This also shows that the uniformity of cementation also improves with the increase of the injection rate. When the injection rate is less than the reaction rate, carbonate is rapidly deposit and then plugs the pores, so that no carbonate precipitate far from the grouting port. When the injection rate is greater than the reaction rate, the solute fills the entire sand column first and then starts to react to form a more uniform carbonate distribution. Fig 10(C) summarized the effect of injecting rate on the suspended bacteria, the attached bacteria, the average

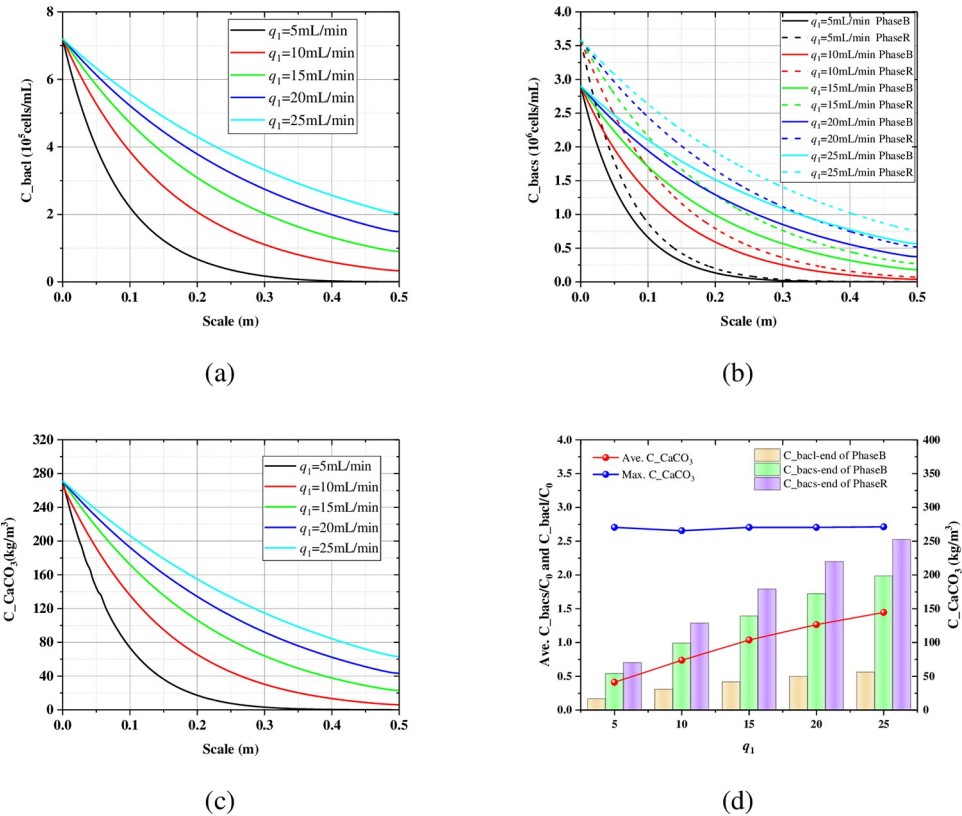

**Fig 10.** The influence of the bacteria injecting rate on the (a) axial profile of the suspended bacteria at the end of Phase B; (b) axial profile of the attached bacteria at the end of the Phase B and Phase C; (c) axial profile of the carbonate at the end of the Phase C; (d) the average bacteria and the carbonate.

carbonate, and the maximum carbonate. With the increase of injecting rate, all the average suspended bacteria, the average attached bacteria and the average carbonate show a significant increasing trend. And it has a more uniform distribution of the calcium carbonate at higher injecting rates, which is very beneficial for ground improvement and load transfer. While the injecting rate has little effect on the maximum carbonate content, the location of which is near the grouting port.

## 4. Applicability and process of the numerical model

Based on this numerical model, the temporal and spatial evolution of various substance (urea, calcium, ammonium, bacteria) concentrations, the porosity and permeability of the ground can be obtained in real time, and more importantly, the influence of the scale size on the simulated results is considered. Different from other MICP numerical models with many parameters, this model can simulate MICP multi-processes with different sand types when three parameters (the constant attachment rate $k_{att}$, the maximum urease constant $u_{sp}$, critical porosity $\varphi_c$) are calibrated. Reliable numerical calculation can provide an important reference for the selection of on-site construction plans and parameters. As shown in Fig 11, when MICP technology is used to reinforce the ground, slopes and banks, it mainly includes the following steps: 1) The basic parameters (porosity, permeability coefficient, scale size) are determined by means of in-situ testing; 2) In order to obtain the optimal parameters (bacterial concentration, injecting rate, injecting time, cement solution concentration) in using MICP technology, the

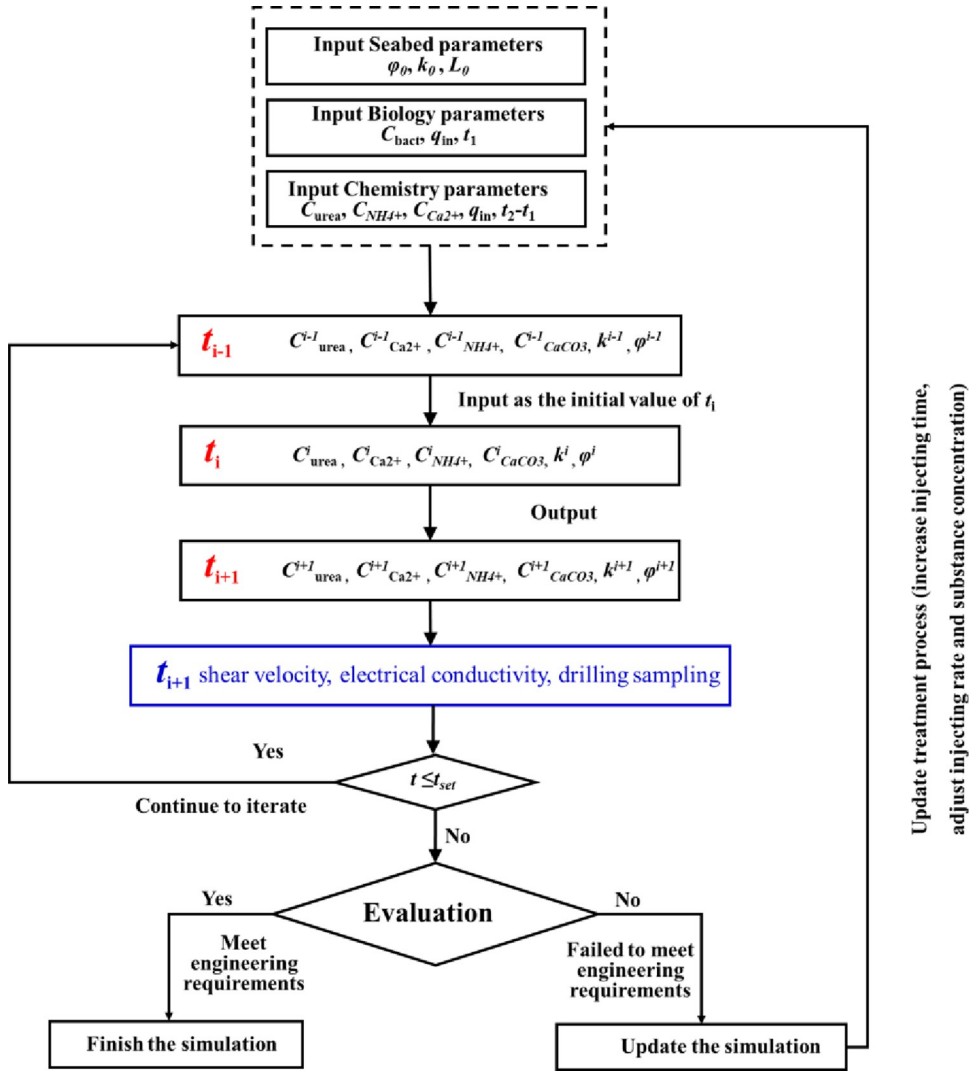

**Fig 11. MICP field reinforcement operation process and numerical simulation.**

sensitivity analysis based on this theoretical model is necessary. Uniform calcium carbonate distribution in MICP reaction is beneficial for load transfer and durability. 3) On-site reinforcement treatment is carried out according to the optimal parameter combination, the strength, stiffness and the uniformity of the ground are evaluated by shear wave velocity, electrical conductivity, drilling sampling and other methods. If the engineering requirements are not met, the secondary reinforcement treatment shall be carried out.

## 5. Conclusions and recommendations

In this paper, the preliminary relationship between scale size and the dispersivity was established, and then the numerical model of the MICP multi-process was optimized. Moreover, the influences of the size effect, initial porosity and the bacterial injecting rate were also analysed.

1. The relationship between the dispersivity and the scale size presented as a substantially linear relationship in the double logarithmic coordinate system. When scale size increases logarithmically from $10^{-2}$ m to $10^{5}$ m, longitudinal dispersivity also shows a trend of increasing

from $10^{-3}$ m to $10^4$ m. The distribution of calcium carbonate is closer to the experimentally measured value when the size effect is considered. According to this relationship, the convection diffusion reaction in the MICP multi-process can be preliminarily simulated.

2. After considering the size effect, the suspended and attached bacteria are higher than the case without considering the size effect, which leads to a higher calcium carbonate content. Scale has little effect on the penetration law of the suspended bacteria. The mean error of the suspended bacteria and the average carbonate content increases with the increase of the scale size.

3. The bacterial injecting rate and initial porosity have significant effects on the distribution of the suspended bacteria, the attached bacteria and the carbonate distribution. The maximum carbonate content increases with the increase of the initial porosity, and the average carbonate showed a significant increasing trend with the increasing of the bacterial injecting rate. It has a more uniform distribution of the calcium carbonate at higher injecting rates or smaller initial porosity, which is very beneficial for ground improvement and load transfer. Both the smaller initial porosity and higher bacterial injecting rate are more conducive to the uniform distribution of carbonate.

4. It is recommended to consider the influence of the scale size on the MICP multi-process in the simulation of the microbial mineralization kinetic model.

5. But it also needs to be pointed out that there are still some problems in this paper. 1) First of all, the microbial model in this model is described by a first-order partial differential equation. In fact, it may be a more complex colloid adsorption distribution, power law distribution, etc. 2) The relationship between dispersivity and the scale size obtained in this article relies on data fitting, so relevant experiments need to be carried out to obtain more data points and increase the degree of fitting.

## Supporting information

**S1 Data. The influence of the scale size.**
(XLSX)

**S2 Data. The influence of the initial porosity.**
(XLSX)

**S3 Data. The influence of the injection rate.**
(XLSX)

## Author Contributions

**Conceptualization:** Xianxian Zhu, Jianhua Wang, Haili Wang, Yujie Li.

**Investigation:** Xianxian Zhu, Jianhua Wang, Yujie Li.

**Methodology:** Jianhua Wang, Yujie Li.

**Writing – original draft:** Xianxian Zhu, Yujie Li.

**Writing – review & editing:** Haili Wang.

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
