## [Decision Letter · Decision Letter 0]

28 Nov 2023

PONE-D-23-20748A numerical model of the MICP multi-process considering the scale sizePLOS ONE

Dear Dr. Li,

Thank you for submitting your manuscript to PLOS ONE. After careful consideration, we feel that it has merit but does not fully meet PLOS ONE’s publication criteria as it currently stands. Therefore, we invite you to submit a revised version of the manuscript that addresses the points raised during the review process.

We look forward to receiving your revised manuscript.

Kind regards,

Abdullah Ekinci, PhD

Academic Editor

PLOS ONE

Journal Requirements:

"The authors would like to acknowledge the supports by the Zhejiang Water Conservancy Key Science and Technology Project (RB2120), Hainan Special PhD Scientific Research Foundation of Sanya Yazhou Bay Science and Technology City (HSPHDSRF-2022-04-002). "

"The authors would like to acknowledge the supports by the Zhejiang Water Conservancy Key Science and Technology Project (RB2120), Hainan Special PhD Scientific Research Foundation of Sanya Yazhou Bay Science and Technology City (HSPHDSRF-2022-04-002)."

"The authors would like to acknowledge the supports by the Zhejiang Water Conservancy Key Science and Technology Project (RB2120), Hainan Special PhD Scientific Research Foundation of Sanya Yazhou Bay Science and Technology City (HSPHDSRF-2022-04-002)."

6. We note that Figures 1 and 2 in your submission contain copyrighted images. All PLOS content is published under the Creative Commons Attribution License (CC BY 4.0), which means that the manuscript, images, and Supporting Information files will be freely available online, and any third party is permitted to access, download, copy, distribute, and use these materials in any way, even commercially, with proper attribution. For more information, see our copyright guidelines: http://journals.plos.org/plosone/s/licenses-and-copyright.

a. You may seek permission from the original copyright holder of Figures 1 and 2 to publish the content specifically under the CC BY 4.0 license. 

Additional Editor Comments:

In general, the paper is well structured, and the data is well analyzed; however, it requires further discussion at a few points. However, I am suggesting the manuscript be accepted for publication if the authors are willing to perform minor improvements/corrections on the submitted work.

Here are the minor improvements / corrections I suggest authors to review:

- Some numerical findings should be included in the abstract.

- Authors should further include research on the introduction section of the paper. Please see the following articles.

- The novelty of this study could be expressed with few sentences at this point.

- Summary and conclusions sections should be re-arranged as Conclusion and Recommendations. In this section limitations and recommendations of this study should be listed.

- There are some of grammatical mistakes and drawbacks in the manuscript, Please improve the English and try to present a concise expression.

- General Comments – Revise the keywords according to journal guidelines.

- General comment – References section should be reviewed as few references are not according to the journal guidelines.

Reviewers' comments:

Reviewer's Responses to Questions

**Comments to the Author**

1. Is the manuscript technically sound, and do the data support the conclusions?

Reviewer #1: Partly

2. Has the statistical analysis been performed appropriately and rigorously? 

Reviewer #1: N/A

3. Have the authors made all data underlying the findings in their manuscript fully available?

Reviewer #1: Yes

4. Is the manuscript presented in an intelligible fashion and written in standard English?

Reviewer #1: No

5. Review Comments to the Author

Reviewer #1: The manuscript need an English language proof reading. There are many grammatical mistakes within the manuscript which has to be corrected specially in Abstract and conclusion part. The abstract has some grammatical issues and awkward sentence structures. For example, the sentence, "Results show that:" can be improved for clarity. Some of the errors could be checked in the lines 10, 12, 14, 17,21, 202, 203, 213, 214, 232, 316, 317, 323

In abstract include specific numerical results or ranges while discussing the findings.

line 53: correct the lab to "laboratory"

Figure 1. This figure is not clear, small fonts and not enough explanation or legends, looks like many different images are just jammed together

Figure 2. The texts in the figures are not readable, it's better to keep only the absolute necessary figures only and give a sort of explanation for each of the figure in caption.

Line 98- In generally, Delete "In"

Figure 3, Y axis legend, dispersivity, small d

Table 2, Sand length? Does it mean sand column/sample length?

Elaborate what are PhaseR, PhaseC, PhaseB, within the text. A space between the word "Phase" and the alphabet after would be needed throughout the manuscript.

Line 195, 196, Avoid parentheses within parentheses, should be checked throughout the text.

Line 300-302: 2) In order to obtain the optimal..... Is this a step in using the MICP technology? Please correct the wordings accordingly

313-314: "Considering the size effect is closer to the experimentally measured value." This sentence is not clear. Better English language use is required.

316 :"higher than that without" what is meant by "that" here?

6. PLOS authors have the option to publish the peer review history of their article (what does this mean?). If published, this will include your full peer review and any attached files.

Reviewer #1: No

---

## [Author Response · Author response to Decision Letter 0]

18 Dec 2023

Responses to Reviewer’s comments

Editor: 

In general, the paper is well structured, and the data is well analyzed; however, it requires further discussion at a few points. However, I am suggesting the manuscript be accepted for publication if the authors are willing to perform minor improvements/corrections on the submitted work.

R. Thank you for the valuable comments and supports to our work. What you mentioned has been revised in the revised manuscript. The following is a point-to-point response to the reviewers’ comments, where letters C & R denote comment and response, respectively. All the modifications have been highlighted in the revised manuscript.

Here are the minor improvements / corrections I suggest authors to review:

-Some numerical findings should be included in the abstract.

R. Thank you for the valuable comments. Some numerical findings were added in the abstract. 

-Authors should further include research on the introduction section of the paper. Please see the following articles.

R. Thank you for the valuable comments. Some researches were added in the introduction section of this paper.

-The novelty of this study could be expressed with few sentences at this point.

R. Thank you for your comments. The novelty of this study were highlighted in the “Abstract” and “Conclusions”.

-Summary and conclusions sections should be re-arranged as Conclusion and Recommendations. In this section limitations and recommendations of this study should be listed.

R. Thank you for your comments. The conclusions sections have been re-arranged as Conclusion and Recommendations. The limitations of this paper were pointed out.

-There are some of grammatical mistakes and drawbacks in the manuscript, Please improve the English and try to present a concise expression.

R. Thank you for your comments. We have reorganized the entire text and revised the grammar and language.

-General Comments – Revise the keywords according to journal guidelines.

R. Thank you for your comments. We have revised the keywords.

-General comment – References section should be reviewed as few references are not according to the journal guidelines.

R. Thank you for your comments. We have revised references in the manuscript.

Reviewer 1’s Comments:

General comments: The manuscript need an English language proof reading. There are many grammatical mistakes within the manuscript which has to be corrected specially in Abstract and conclusion part. 

R. Thank you for your comments and give us a chance to reply. The following is a point-to-point response to your comments. All the modifications have been highlighted in the revised manuscript.

C. The abstract has some grammatical issues and awkward sentence structures. For example, the sentence, "Results show that:" can be improved for clarity. Some of the errors could be checked in the lines 10, 12, 14, 17,21, 202, 203, 213, 214, 232, 316, 317, 323. In abstract include specific numerical results or ranges while discussing the findings.

R. Thank you for the valuable comments. The errors in the lines 10, 12, 14, 17,21, 202, 203, 213, 214, 232, 316, 317, 323 were revised in the manuscript. The specific numerical results or ranges were added in abstract and conclusions.

C. line 53: correct the lab to "laboratory"

R. The “lab” was corrected to “laboratory” in the revised manuscript.

C. Figure 1. This figure is not clear, small fonts and not enough explanation or legends, looks like many different images are just jammed together

C. Figure 2. The texts in the figures are not readable, it's better to keep only the absolute necessary figures only and give a sort of explanation for each of the figure in caption.

R. To avoid causing confusion and copyright issues, Fig. 1 and Fig. 2 have been deleted in the revised manuscript.

C. Line 98- In generally, Delete "In"

R. The “In generally” was corrected to “In general” in the revised manuscript.

C. Figure 3, Y axis legend, dispersivity, small d

R. This picture has been modified according to your comment.

C. Table 2, Sand length? Does it mean sand column/sample length?

R. Thank you for the valuable comments. The “sand length” in Table 2 was revised to “sand sampler length” in the revised manuscript.

C. Elaborate what are PhaseR, PhaseC, PhaseB, within the text. A space between the word "Phase" and the alphabet after would be needed throughout the manuscript.

R. Thank you for the valuable comments. A space between the word "Phase" and the alphabet after was added in the revised manuscript.

C. Line 195, 196, Avoid parentheses within parentheses, should be checked throughout the text.

R. The parentheses within parentheses were revised.

C. Line 300-302: 2) In order to obtain the optimal..... Is this a step in using the MICP technology? Please correct the wordings accordingly

C. Line 300-302: 2) In order to obtain the optimal..... Is this a step in using the MICP technology? Please correct the wordings accordingly

R. This sentence was revised to “In order to obtain the optimal parameters (bacterial concentration, injecting rate, injecting time, cement solution concentration) in using MICP technology, the sensitivity analysis based on this theoretical model is necessary. Uniform calcium carbonate distribution in MICP reaction is beneficial for load transfer and durability”. 

C. 313-314: "Considering the size effect is closer to the experimentally measured value." This sentence is not clear. Better English language use is required.

R1. This sentence was revised to “The distribution of calcium carbonate is closer to the experimentally measured value when the size effect is considered”.

C. 316 :"higher than that without" what is meant by "that" here?

R. Thank you for your valuable comments. This sentence was revised to “2) After considering the size effect, the suspended and attached bacteria are higher than the case without considering the size effect, which leads to a higher calcium carbonate content”.

---

## [Editor Report · Decision Letter 1]

2 Jan 2024

A numerical model of the MICP multi-process considering the scale size

PONE-D-23-20748R1

Dear Dr. Li,

We’re pleased to inform you that your manuscript has been judged scientifically suitable for publication and will be formally accepted for publication once it meets all outstanding technical requirements.

Kind regards,

Abdullah Ekinci, PhD

Academic Editor

PLOS ONE
---

## [Editor Report · Acceptance letter]

14 Jan 2024

PONE-D-23-20748R1 

PLOS ONE

Dear Dr. Li, 

I'm pleased to inform you that your manuscript has been deemed suitable for publication in PLOS ONE. Congratulations! Your manuscript is now being handed over to our production team.

Kind regards, 

on behalf of

Assoc. Prof. Dr. Abdullah Ekinci 

Academic Editor

PLOS ONE